# Real world clinical practice in treating advanced hepatocellular carcinoma: When East meets West

Yi-Hao Yen[1], Yu-Fan Cheng[2], Jing-Houng Wang[1], Chih-Che Lin[3], Yen-Yang Chen[4], Chee-Chien Yong[3], Yueh-Wei Liu[3], Jen-Yu Cheng[5], Chien-Hung Chen[1], Tsung-Hui Hu[1]*

1 Division of Hepatogastroenterology, Department of Internal Medicine, Kaohsiung Chang Gung Memorial Hospital and Chang Gung University College of Medicine, Kaohsiung, Taiwan, 2 Liver Transplantation Center, Department of Diagnostic Radiology, Kaohsiung Chang Gung Memorial Hospital, Chang Gung University College of Medicine, Kaohsiung, Taiwan, 3 Liver Transplantation Center and Department of Surgery, Kaohsiung Chang Gung Memorial Hospital, Kaohsiung, Taiwan, 4 Division of Hematology-Oncology, Department of Internal Medicine, Kaohsiung Chang Gung Memorial Hospital, Kaohsiung, Taiwan, 5 Department of Radiation Oncology, Kaohsiung Chang Gung Memorial Hospital, Kaohsiung, Taiwan

* dr.hu@msa.hinet.net

**Data Availability Statement:** All relevant data are within the paper and its Supporting Information files.

## Abstract

### Background and aims

The Barcelona Clinic Liver Cancer (BCLC) stage C (BCLC C) of hepatocellular carcinoma (HCC) includes a heterogeneous population for which sorafeninb is one of the recommended therapies. We aim to evaluate the real world clinical treatment and survival of BCLC stage C patients in an Asian cohort.

### Methods

This is a retrospective cohort study that enrolled 427 consecutive BCLC stage C patients diagnosed between 2011 and 2017 by using the HCC registry data for our hospital. All patients were managed via a multidisciplinary team (MDT) approach.

### Results

Hepatitis B surface antigen positive was noted in 50.6% of the patients. The patients were classified as performance status (PS)1 alone (n = 83; 19.4%), PS2 alone (n = 23; 5.4%), or macrovascular invasion (MVI) or extrahepatic spread (EHS) (n = 321; 75.2%). The median overall survival (OS) was 11.0 months in the whole cohort. The most frequent treatments were transcatheter arterial embolization (TAE) in the PS1 (45.8%) and PS2 patients (52.2%) and sorafenib (32.4%) in the MVI or EHS patients. The independent prognostic factors were the PS, Child-Pugh class, MVI or EHS, alpha fetoprotein levels, and treatment type.

### Conclusions

We reported the real world management in BCLC stage C patients in an Asian cohort through the use of personalized management via a MDT approach.

**Funding:** Financial support: This study was supported by Grant CMRPG8J1281 from the Kaohsiung Chang Gung Memorial Hospital, Taiwan. Grant Recipient is Yi-Hao Yen. The funders had no role in study design, data collection and analysis, decision to publish, or preparation of the manuscript.

**Competing interests:** The authors have declared that no competing interests exist

## Introduction

Hepatocellular carcinoma (HCC) is the third most frequent cause of cancer death in the world [1, 2]. HCC is the first and second leading cause of cancer-related mortality in males and females, respectively, in Taiwan [3, 4].

Barcelona Clinic Liver Cancer (BCLC) stage C patients are a heterogeneous population that includes patients with tumors that have macroscopic vascular invasion (MVI), patients with tumors that have extrahepatic spread (EHS), and patients with mild cancer-related symptoms (that is those with the Eastern Cooperative Oncology Group [ECOG] Performance Status [PS] grades 1–2) [5]. Sorafenib is one of the recommended treatments for BCLC stage C patients [6], which is nonetheless suboptimal in efficacy and tolerability [7, 8]. A recent Italian study reported real world clinical practice results in treating naïve BCLC stage C HCC [9]. They found that the survival duration of BCLC stage C patients was improved through the use of personalized management via a multidisciplinary team (MDT) approach. However, the leading etiology of chronic liver disease in the European population is hepatitis C virus (HCV), and validation of the Italian study's approach in other populations with different ethnic and clinical backgrounds, as well as in contexts with variations in the provision of systemic and local-regional therapies, is needed.

We therefore investigated the real world clinical treatment and survival of BCLC stage C patients in an Asian medical center, most of whom were infected with hepatitis B virus (HBV).

## Patients and methods

This is a retrospective study using the last version of the Kaohsiung Chang Gung Memorial Hospital HCC registry data, and consisted of data for 427 BCLC stage C HCC patients consecutively evaluated and managed from January 2011 to December 2017 at this hospital. A flow chart of the patients' enrollment is shown in Fig 1. The data were prospectively collected and updated every 2 years. We contacted any patients who were lost to follow-up by phone and checked their vital status using the Cancers Screening and Tracing Information Integrated System for Taiwan (https://hosplab.hpa.gov.tw/CSTIIS/index.aspx). The HCC registry data was managed by Ms. Hui Ping Tseng, who has performed this work since 2005. The definition of HCC was based on assessment via an MDT conference and/or on international guidelines [10–14]. HCC diagnosis was histological in 240 (56.2%) cases, and in the remaining cases, it was based on typical features in imaging studies.

The patients were classified into three groups according to the characteristics by which a patient is determined to have BCLC stage C HCC. That is, they were grouped into PS1 alone, PS2 alone, and MVI or EHS groups. The PS of each patient was assessed according to the standards of the ECOG [5]. The cancer registry data recorded the first-line treatment or treatments for the patients, which included the following: liver transplant, resection, radiofrequency ablation (RFA), transcatheter arterial chemoembolization/embolization (TACE/TAE), sorafenib, systemic chemotherapy, hepatic artery infusion chemotherapy (HAIC), external beam radiation therapy (EBRT), and/or best supportive care (BSC).

Sorafenib treatment has been reimbursed by the National Health Insurance Administration (NHIA) in Taiwan since August 2012 for patients with portal vein tumor thrombus (PVTT) (Vp3 or Vp4) [15] or EHS and Child-Pugh class A liver disease. Therefore, BCLC C patients with PS1 alone, PS2 alone, Child-Pugh class B liver disease, peripheral PVTT (Vp1 or Vp2), hepatic veins tumor thrombus, inferior vena cava tumor thrombus and those diagnosed during 2011 to August 2012 were not reimbursed.

For patients with PS1 alone or PS2 alone, the criteria for allocating patients to the different therapeutic approaches are as follows:

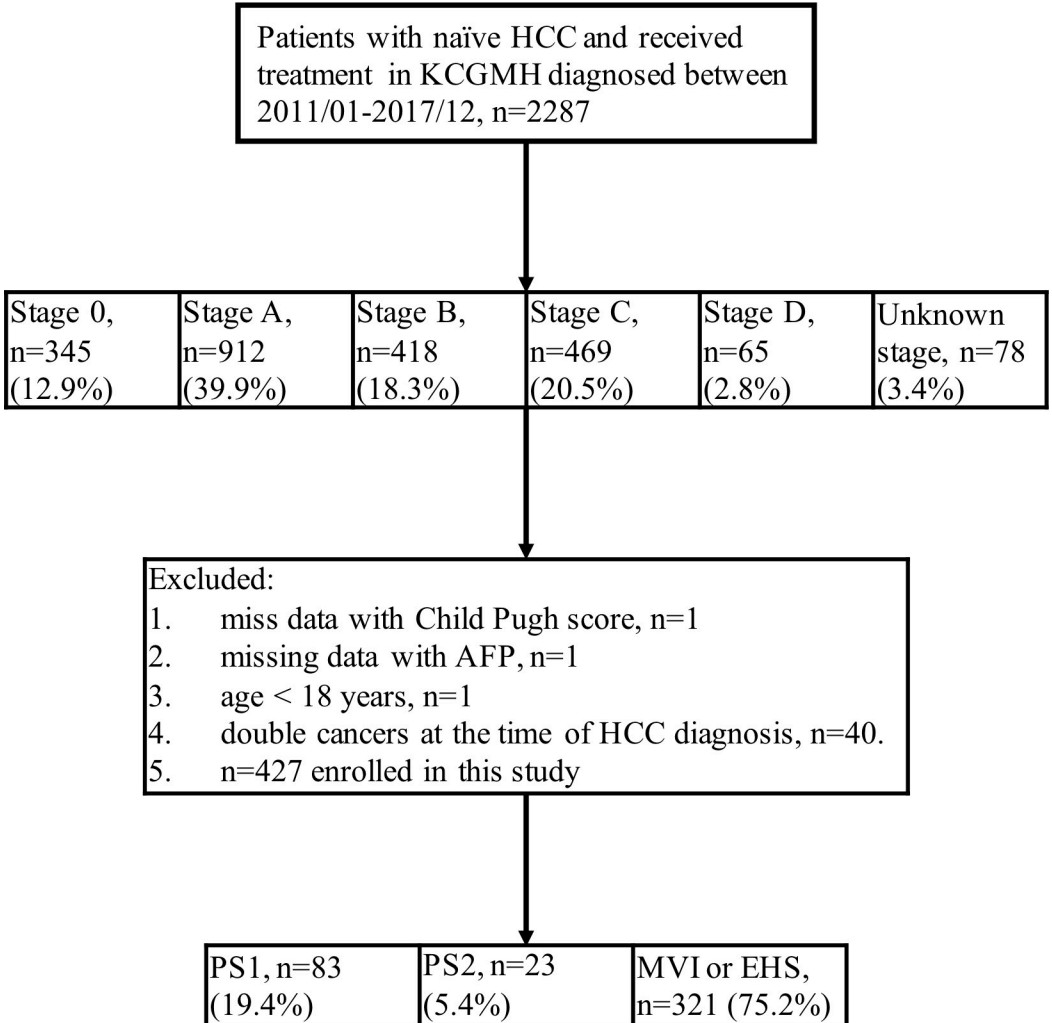

**Fig 1. Flow chart of the patient enrollment from the Kaohsiung Chang Gung Memorial Hospital (KCGMH) cancer database.** HCC, hepatocellular carcinoma; HCC was staged according to the Barcelona Clinic Liver Cancer staging system. AFP, Alpha fetoprotein; PS, Performance status; MVI, macrovascular invasion; EHS, extrahepatic spread.

1. Liver transplant is recommended as the first-line option for HCC within the University of California at San Francisco (UCSF) criteria but unsuitable for resection [16].

2. The general principle for liver resection of HCC followed the recommendations of the European Association for the Study of the Liver (EASL) guidelines [17], which are based on a multi-parametric composite assessment of liver function, the extent of resection, the future liver remnant, PS, and the patient's comorbidities. Liver resection is indicated in HCC patients with tumors located in one lobe of the liver [18].

3. RFA is indicated for patients with Child-Pugh class A or B liver disease and HCC within Milan criteria [18].

4. TACE or EBRT is indicated for patients with Child-Pugh class A or B liver disease and HCC within Milan criteria [10] in whom it is difficult to perform RFA because of medical comorbidities or tumor location [19].

5. TACE is indicated for patients with Child-Pugh class A or B liver disease and tumor burden at the BCLC stage B [19].

6. EBRT is indicated for patients with Child-Pugh class A or B liver disease and tumor burden at the BCLC stage B unsuitable to TACE [19].

Under most circumstances, patients with MVI or EHS were treated with sorafenib if they met the criteria of reimbursement. The criteria for allocating patients to the different therapeutic approaches are as follows:

1. RFA is indicated for patients with Child-Pugh class A or B liver disease and intrahepatic tumor burden within Milan criteria [10], with small metastatic lymph nodes or tiny lung nodules, misclassified as benign at the time of treatment.

2. TAE/TACE is indicated for patients with Child-Pugh class A or B liver disease and intrahepatic tumor burden at the BCLC stage B, with small metastatic lymph nodes or tiny lung nodules, misclassified as benign at the time of treatment.

3. Systemic chemotherapy or HAIC is indicated in patients with Child-Pugh class A liver disease and adequate hemogram data (white blood cell $\geq$ 4000/mL, platelet $\geq$ 100,000/mL) [19].

4. HAIC is indicated for patients with MVI [19].

5. Systemic chemotherapy is indicated for patients with EHS [19].

6. Liver resection is indicated in patients with HCC involving the ipsilateral portal vein. Resection is contraindicated in patients with HCC with contralateral portal vein or portal vein main trunk invasion [19].

7. Liver resection is indicated in patients with HCC involving ipsilateral hepatic vein. Liver resection is contraindicated in patients with HCC with right atrium or inferior vena cava involvement [19].

8. Liver resection is indicated in patients with HCC with extra-hepatic, single organ involvement. Liver resection is contraindicated in patients with HCCs with extra-hepatic, multiple organ involvement [19].

9. EBRT is indicated in patients with MVI and Child-Pugh class A or B liver disease [19].

## The standard procedure of the MDT in our center

The MDT in our center includes hepatologists, radiologists, pathologists, transplant surgeons, surgical oncologists, radiation oncologists, medical oncologists, and nurses. The leader of our MDT is Yu-Fan Cheng. The MDT meetings are held every Tuesday, and all new HCC cases are discussed. Treatment experiences cases are discussed if the physician in charge requests. If MDT recommends BSC, the physician in charge will consult the palliative care professionals. If liver transplant is recommended, we refer these patients to liver transplant MDT for detailed discussion.

Our MDT include two nurses who administer and coordinate the meetings. The general flow begins with a brief clinical presentation by the physician in charge, followed by images review, discussion, and decision. A standardized one-page form is used. This form records the relevant clinical and laboratory data that determine liver function, PS, the tumor stage and the consensus recommendations. The physician in charge informs patients of the MDT recommendations. The document of the MDT recommendations is kept in the patient's medical chart.

This study was approved by the Institutional Review Board of Kaohsiung Chang Gung Memorial Hospital (IRB number: 201901120B0). The requirement for informed consent was waived by the IRB.

## Statistical analysis

Continuous variables were summarized as median and interquartile range, while categorical variables were summarized as frequency and relative percentage. Comparisons of continuous variables were carried out using the one-way ANOVA test, and comparisons between categorical variables were carried out using the Fisher's exact test or the chi-squared test, as appropriate. Patient survival was calculated from the date of HCC diagnosis to the date of death or last contact. Overall survival (OS) was estimated using the Kaplan-Meier method, and the differences between groups were estimated with the log-rank test. Cox proportional hazard regression analysis was used to evaluate the risk factors for mortality. Covariates in the multivariable model were chosen a priori for clinical importance. Potential confounders included age, gender, ECOG PS, the presence of MVI or EHS, Alpha fetoprotein (AFP), and treatment. These variables were always retained in the model in the multivariate regression analysis. A *P* value less than 0.05 in a two-tailed test was considered statistically significant. All analyses were performed using Stata version 14.0 (StataCorp 2015 Stata Statistical Software: Release 14. College Station, TX: StataCorp LP.).

## Results

### Patients' characteristics

Table 1 shows the demographic and clinical features of the BCLC stage C HCC patients classified according to PS and the presence of MVI or EHS. PS1 patients accounted for 19.4% of the entire cohort (n = 83), PS2 patients accounted for 5.4% (n = 23), and MVI or EHS patients accounted for 75.2% (n = 321). The patients in the MVI or EHS subgroup were younger and had a lower average creatinine level than the patients in the other two subgroups. They also included higher proportions of male and hepatitis B surface antigen (HBsAg)-positive alone patients and a lower proportion of portal hypertension (defined as portal systemic collateral or splenomegaly in image studies) patients. As expected, the proportions of patients with multiple tumors, tumor size> 5 cm, and AFP > 200 ng/dL were higher in the MVI or EHS subgroup.

### Treatment

The treatment distributions differed significantly among the BCLC C subgroups (Table 2). In the PS1 and PS2 patients, TAE/TACE was most frequently applied (45.8% in the PS1 subgroup, 52.2% in the PS2 subgroup), followed by curative therapies, whereas sorafenib was not administered to any of the patients in the PS1 or PS2 subgroups. As expected, among the MVI or EHS patients, sorafenib was the most frequent treatment (32.4%), followed by various other treatments (i.e., systemic chemotherapy, HAIC, or EBRT) in 29.6% of the patients.

### Survival analyses

During the median follow-up period of 10.0 months [95% confidence interval (CI), 8.0–12.0], 296 (69.3%) patients died. The numbers of deaths across the subgroups were as follows: 33 (39.8%) in the PS1 group, 12 (52.2%) in the PS2 group, and 251 (78.2%) in the MVI or EHS group. In the whole population, the median OS was 11.0 months (95% CI, 9.0–13.0). The survival duration differed across the BCLC stage C subgroups (P < 0.001; Fig 2). It was 22.0 months (95% CI, 19.0–27.0) in the PS1 patients, 10.0 months (95% CI, 4.0–22.0) in the PS2

Table 1. Demographic and clinical characteristics of the patients with advanced (BCLC C) HCC (N = 427).

| Variables | | PS1 (n = 83) | PS2 (n = 23) | MVI or EHS (n = 321) | P |
|---|---|---|---|---|---|
| Age | Years | 67 (60–78) | 69 (55–81) | 60 (51–68) | <0.001 |
| Sex | Male | 55 (66.3%) | 13 (56.5%) | 267 (83.2%) | <0.001 |
| Etiology | HBsAg positive alone | 24 (28.9%) | 3 (13.0%) | 123 (38.3%) | 0.002 |
| | Anti-HCV positive alone | 23 (27.7%) | 11 (47.8%) | 57 (17.8%) | |
| | Alcohol abuse alone | 4 (4.8%) | 1 (4.3%) | 13 (4.0%) | |
| | Mixed = any combination | 9 (10.8%) | 3 (13.0%) | 69 (21.5%) | |
| | all negative | 23 (27.7%) | 5 (21.7%) | 59 (18.4%) | |
| ECOG PS | 0 | 0 (0%) | 0 (0%) | 207 (64.5%) | <0.001 |
| | 1 | 83 (100.0%) | 0 (0%) | 82 (25.5%) | |
| | 2 | 0 (0%) | 23 (100.0%) | 32 (10.0%) | |
| Bilirubin | mg/dL | 1.1 (0.8–1.8) | 1.0 (0.8–2.3) | 1.2 (0.9–1.9) | 0.89 |
| Creatinine | mg/dL | 1.2 (0.9–1.9) | 1.3 (0.8–1.7) | 1.0 (0.8–1.2) | <0.001 |
| INR | | 1.1 (1.0–1.2) | 1.1 (1.0–1.3) | 1.1 (1.0–1.2) | 0.84 |
| Child Pugh class | A | 59 (71.1%) | 14 (60.9%) | 237 (73.8%) | 0.38 |
| | B | 24 (28.9%) | 9 (39.1%) | 84 (26.2%) | |
| Portal hypertension | Yes | 42 (50.6%) | 17 (73.9%) | 141 (43.9%) | 0.02 |
| Tumor number | Single | 49 (59.0%) | 16 (69.6%) | 124 (38.6%) | <0.001 |
| | Multiple | 34 (41.0%) | 7 (30.4%) | 197 (61.4%) | |
| Tumor size | <5cm | 50 (60.2%) | 18 (78.3%) | 55 (17.1%) | <0.001 |
| | ≥5cm | 32 (38.6%) | 4 (17.4%) | 255 (79.4%) | |
| Ascites | Yes | 24 (28.9%) | 8 (34.8%) | 66 (20.6%) | 0.10 |
| AFP>200 ng/dL | Yes | 20 (24.1%) | 8 (34.8%) | 205 (63.9%) | <0.001 |
| Diagnosis | Pathology | 46 (55.4%) | 7 (30.4%) | 187 (58.3%) | 0.03 |
| | Clinical | 37 (44.6%) | 16 (69.6%) | 134 (41.7%) | |

HBsAg, hepatitis B surface antigen; HCV, hepatitis C virus; ECOG, Eastern Cooperative Oncology Group; PS, Performance status; INR, International Normalized Ratio; AFP, Alpha fetoprotein; MVI, macrovascular invasion; EHS, extrahepatic spread; BCLC, Barcelona Clinic Liver Cancer

patients, and 7.0 months (95% CI, 6.0–8.6) in the MVI or EHS patients. The PS1 patients had a significantly longer survival duration than the PS2 patients (P = 0.024). The survival duration was not significantly different for the MVI or EHS patients compared to the PS2 patients (P = 0.672).

Table 2. Treatment distribution in the various BCLC C subclasses.

| | PS1 (n = 83) | PS2 (n = 23) | MVI or EHS (n = 321) |
|---|---|---|---|
| Transplant | 5 (6.0%) | 3 (13.0%) | 0 (0%) |
| Resection | 20 (24.1%) | 1 (4.3%) | 55 (17.1%) |
| RFA | 13 (15.7%) | 6 (26.1%) | 6 (1.9%) |
| TAE/TACE | 38 (45.8%) | 12 (52.2%) | 55 (17.1%) |
| Sorafenib | 0 (0%) | 0 (0%) | 104 (32.4%) |
| Other | 7 (8.4%) | 1 (4.3%) | 95 (29.6%) |
| BSC | 0 (0%) | 0 (0%) | 6 (1.9%) |

Other treatment (i.e., systemic chemotherapy, hepatic artery infusion chemotherapy or external beam radiation therapy). RFA, radiofrequency ablation; TACE/TAE, transcatheter arterial chemoembolization/embolization; BSC, best supportive care; PS, Performance status; MVI, macrovascular invasion; EHS, extrahepatic spread; BCLC, Barcelona Clinic Liver Cancer

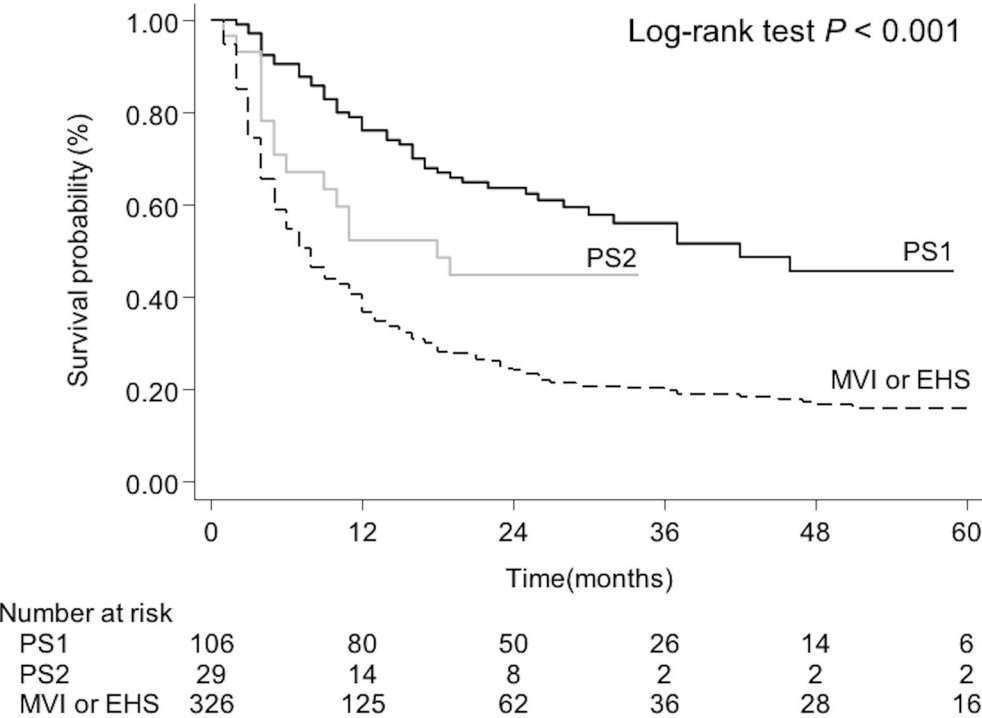

**Fig 2. Survival of patients according to BCLC C subclasses.** PS, Performance status; MVI, macrovascular invasion; EHS, extrahepatic spread.

To assess the degree of MVI on the effect of OS, MVI patients were divided further according to the location/extent of vascular invasion. According to the Hong Kong Liver Cancer (HKLC) staging system [20], intrahepatic MVI was defined as tumor invasion of the intrahepatic branches of the portal vein or hepatic veins. Extrahepatic MVI was defined as tumor invasion of the main portal trunk or inferior vena cava.

There were 321 patients with MVI or EHS in our study. Among these, 75 patients had MVI +EHS (Patients with both MVI and EHS), 51 had EHS alone (Patients with EHS and without MVI), and 195 had MVI alone (Patients with MVI and without EHS). For patients with MVI alone (n = 195), according to the HKLC staging system [20], there were 73 with extrahepatic vascular invasion [i.e., central MVI (c-MVI)], 120 with intrahepatic vascular invasion [i.e., peripheral MVI (p-MVI)], and 2 with unclassified MVI (i.e., the location/extent of PVT was not mentioned in the image reports). The median OS was longer in patients with p-MVI (18.0 months; 95% CI, 12.0–24.0) compared to those with c-MVI (6.0 months; 95% CI, 3.0–7.0; p<0.001; Fig 3).

After patients with hepatic vein and/or inferior vena cava tumor invasion (n = 48) were excluded, there are 58 patients with central (portal trunk) PVTT, 87 with peripheral PVTT, and 2 with unclassified PVTT. The median OS was longer in patients with p-MVI (14.1 months; 95% CI, 9.0–22.0) compared to those with c-MVI (5.1 months; 95% CI, 3.0–7.0; p = 0.002; Fig 4).

Fifty-five (17.1%) of the patients with MVI or EHS in this study underwent surgical resection. Among those, 2 had MVI and EHS, 5 had EHS alone, and 48 had MVI alone. The median overall survival was 67 months; 95% CI, 23-not available (not available indicates that the 95% CI survival has not yet been reached).

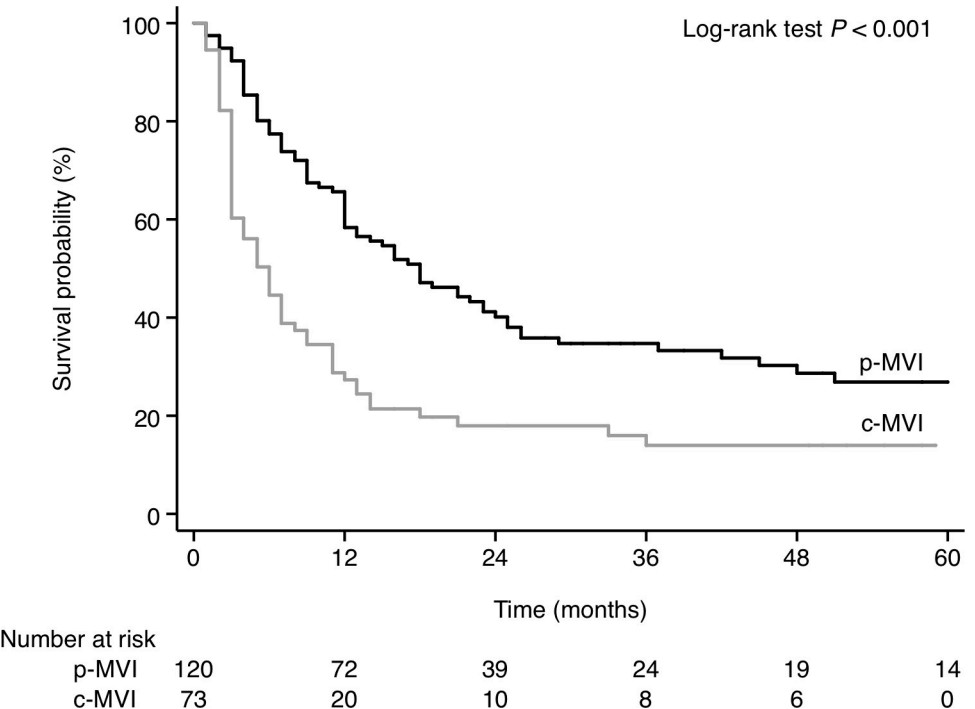

**Fig 3. Survival of patients according to the type of MVI extension (black line, p-MVI; gray line, c-MVI).** p-MVI was defined as tumor invasion of the intrahepatic branches of the portal vein or hepatic veins. c-MVI was defined as tumor invasion of the main portal trunk or inferior vena cava.

## Prognostic factors

The univariate analysis results regarding prognostic factors associated with mortality are shown in Table 3. This analysis showed that mortality was associated with age < 70 years, ECOG PS = 2, Child-Pugh class B, multiple tumors, MVI or EHS, tumor size > 5cm, AFP > 200 ng/dL, bilirubin > 1.1 mg/dL, international normalized ratio (INR) > 1.25, and treatment. Multivariate regression analysis showed that ECOG PS 2, Child-Pugh class B, MVI or EHS, AFP >200 ng/mL, and treatment were independently associated with mortality (Table 4).

## Discussion

Previous studies have reported that adherence to BCLC guidelines was poor in BCLC C patients [9, 21, 22]. Poor adherence among these patients may be due to some of them being treated with a single drug, sorafenib, which is suboptimal in tolerability and efficacy. Furthermore, the low adherence of such patients to the guidelines may be supported by studies showing that, in selected cases, better results were noted with local-regional therapies and surgery [23, 24].

Of the PS1 patients in this study, more than 90% were treated with curative therapies or TAE/TACE, and no one received sorafenib treatment. PS is related to cancer-related symptoms. However, 59% of patients had single tumor and 60% of patients had tumors less than 5 cm in this group. These patients with a low tumor burden might not have cancer-related symptoms. However, Giannini, et al reported that the attribution of a cancer-dependent—as recommended by the BCLC system—mild deterioration of PS is very subjective. Several

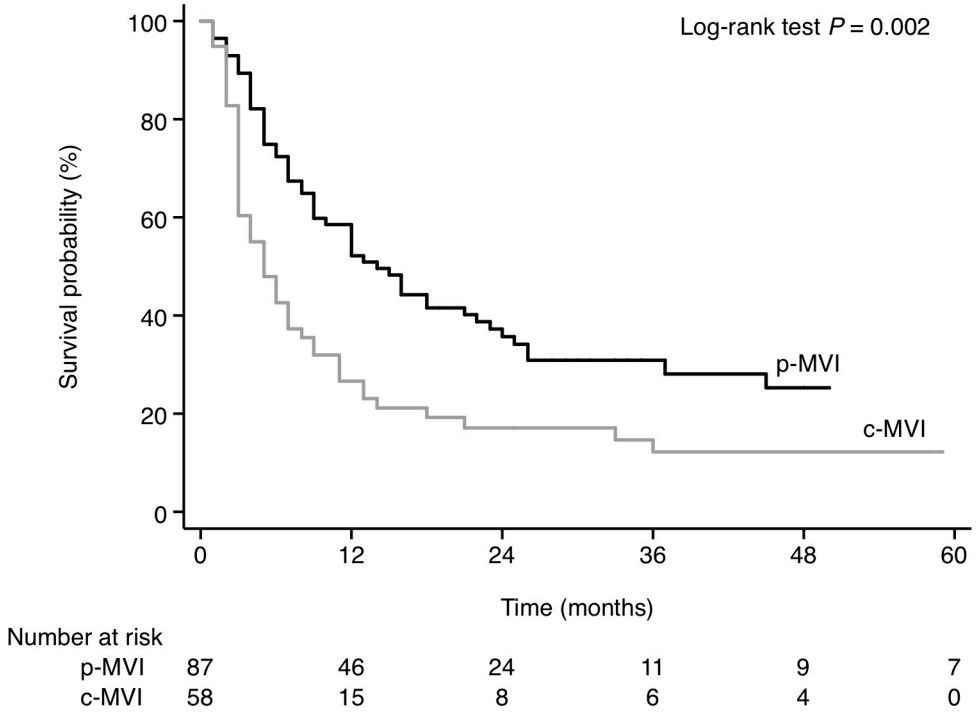

**Fig 4. Survival of patients according to the type of MVI extension (black line, p-MVI; gray line, c-MVI).** c-MVI was defined as central (portal trunk) portal vein tumor thrombus (PVTT). p-MVI was defined as peripheral PVTT.

confounding factors, such as old age, presence of decompensated cirrhosis and the extrahepatic comorbidities may play a major role [9].

The American Association for the Study of Liver Diseases (AASLD) guidelines recently revised the BCLC staging [6]. The PS for BCLC stages 0, A, and B has been changed to 0–1 to better reflect clinical practice due to the significant overlap that exists between PS0 and PS1 and the potential bias of physician and patient-reported PS [25].

For the PS2 patients in this study, the most common treatment was TAE/TACE (52.2%), while none of them received sorafenib treatment. The survival of these patients was not significantly different from that of the patients with MVI or EHS, which may have been due to the small number of patients in the PS2 alone subgroup.

For the MVI or EHS patients in this study, sorafenib was the most frequently administered treatment (32.4% of cases). However, 17.1% of the patients with MVI or EHS in this study underwent surgical resection. Median OS was 67 months in these patients. According to western guidelines, HCC with MVI is a contraindication for liver resection [6, 17]. In contrast, liver resection is recommended for selected patients with MVI or EHS in Taiwan [19].

Nearly 30% of the patients with MVI or EHS in our cohort underwent other treatments (i.e., systemic chemotherapy, HAIC, or EBRT). In contrast, none of the patients in the Italian study received these kinds of treatment [9]. In the west, although EBRT is being increasingly used in patients with MVI, it is not recommended by the EASL guideline [17]. In contrast, EBRT is recommended for patients with HCC and MVI in the Asia-Pacific and Taiwan guidelines [19, 26]. Meanwhile, HAIC is widely used for patients with HCC and PVTT in Japanese hospitals [27]. It is also one of the treatment modalities for patients with MVI recommended by the Taiwan guideline [19].

**Table 3. Risk factors for mortality in patients with advanced HCC (BCLC C Stage).**

| Variables | Univariate HR (95% CI) | *P* value |
|---|:---:|:---:|
| Age (years) | | |
| < = 70 | 1 | |
| >70 | 0.69 (0.53–0.91) | **0.008** |
| Sex | | |
| Female | 1 | |
| Male | 1.18 (0.88–1.59) | 0.26 |
| Etiology | | |
| Anti-HCV positive alone | 1 | |
| HBs Ag positive alone | 0.92 (0.67–1.25) | 0.58 |
| Alcohol abuse alone | 1.12 (0.60–2.07) | 0.73 |
| Mixed | 1.13 (0.79–1.61) | 0.51 |
| Others | 0.85 (0.59–1.21) | 0.37 |
| ECOG PS | | |
| 0–1 | 1 | |
| 2 | 1.48 (1.05–2.08) | **0.03** |
| Child Pugh class | | |
| A | 1 | |
| B | 1.88 (1.46–2.42) | **<0.001** |
| Portal hypertension | | |
| No | 1 | |
| yes | 0.91 (0.72–1.14) | 0.40 |
| Ascites | | |
| No | 1 | |
| yes | 1.26 (0.96–1.65) | 0.10 |
| Tumor number | | |
| Single | 1 | |
| Multiple | 1.99 (1.57–2.53) | **<0.001** |
| Subgroup | | |
| PS1+PS2 | 1 | |
| MVI or EHS | 2.77 (2.01–3.82) | **<0.001** |
| Tumor size (cm) | | |
| <5 | 1 | |
| > = 5 | 1.88 (1.43–2.48) | **<0.001** |
| AFP (ng/dL) | | |
| < = 200 | 1 | |
| >200 | 1.78 (1.41–2.26) | **<0.001** |
| Bilirubin (mg/dL) | | |
| < = 1.1 | 1 | |
| >1.1 | 1.36 (1.08–1.71) | **0.01** |
| INR | | |
| <1.25 | 1 | |
| > = 1.25 | 1.54 (1.21–1.96) | **0.001** |
| Creatinine (mg/dL) | | |
| < = 1.2 | 1 | |
| >1.2 | 1.11 (0.86–1.43) | 0.43 |
| Treatment | | |
| Curative (transplant + resection + RFA) | 1 | |

(*Continued*)

**Table 3.** (Continued)

| Variables | Univariate HR (95% CI) | P value |
|---|---|---|
| TACE/TAE | 3.02 (2.04–4.47) | <0.001 |
| Sorafenib | 5.85 (3.98–8.60) | <0.001 |
| BSC | 16.09 (6.69–38.69) | <0.001 |
| others | 4.86 (3.29–7.19) | <0.001 |

HBsAg, hepatitis B surface antigen; HCV, hepatitis C virus; ECOG, Eastern Cooperative Oncology Group; PS, Performance status; INR, International Normalized Ratio; AFP, Alpha fetoprotein; MVI, macrovascular invasion; EHS, extrahepatic spread; BCLC, Barcelona Clinic Liver Cancer; RFA, radiofrequency ablation; TACE/TAE, transcatheter arterial chemoembolization/embolization; BSC, best supportive care

Few of the patients with EHS in this study underwent systemic chemotherapy, and these patients were all diagnosed before the initiation of sorafenib treatment reimbursement by the NHIA.

Our study showed that the OS of peripheral MVI patients was significantly longer than in the central MVI patients, similar finding was noted in the publication by Giannini et al [9].

Serper et al. reported that subspecialist care within 30 days of HCC diagnosis and review by an MDT were associated with reduced mortality compared with care provided by gastroenterologists

**Table 4. Independent risk factors for mortality in patients with advanced HCC (multivariate regression analysis).**

| Variables | Multivariate HR (95% CI) | P value |
|---|---|---|
| Age (years) | | |
| ≤ 70 | 1 | |
| > 70 | 0.79 (0.6–1.04) | 0.10 |
| Sex | | |
| Female | 1 | |
| Male | 0.82 (0.61–1.11) | 0.20 |
| ECOG PS | | |
| 0–1 | 1 | |
| 2 | 1.66 (1.19–2.33) | **0.003** |
| Child Pugh class | | |
| A | 1 | |
| B | 1.90 (1.47–2.46) | <0.001 |
| Subgroup | | |
| PS1+PS2 | 1 | |
| MVI or EHS | 2.15 (1.55–3.00) | <0.001 |
| AFP (ng/dL) | | |
| ≤ 200, as reference | 1 | |
| > 200 | 1.36 (1.07–1.73) | **0.01** |
| Treatment | | |
| Curative (transplant + resection + RFA) | 1 | |
| TAE | 2.81 (1.94–4.08) | <0.001 |
| Sorafenib | 4.25 (2.92–6.19) | <0.001 |
| BSC | 9.35 (3.87–22.57) | <0.001 |
| others | 3.32 (2.27–4.85) | <0.001 |

ECOG, Eastern Cooperative Oncology Group; PS, Performance status; INR, International Normalized Ratio; AFP, Alpha fetoprotein; MVI, macrovascular invasion; EHS, extrahepatic spread; BCLC, Barcelona Clinic Liver Cancer; RFA, radiofrequency ablation; TACE/TAE, transcatheter arterial chemoembolization/embolization; BSC, best supportive care; AFP, Alpha fetoprotein

in a community setting in a Veterans Administration cohort [28]. The current healthcare system in Taiwan, known as the National Health Insurance system, was instituted in 1995. The population coverage reached and has been maintained at or above 99% since 2004. Under the system, citizens are free to choose hospitals and physicians without referral. The NHIA offers reimbursement for all treatment modalities for HCC patients, as patients with cancers can apply a catastrophic illness card. As such, patients with HCC do not have to pay anything when they receive medical care related to HCC. Furthermore, the island of Taiwan has an area of 35,808 square kilometers, which is smaller than that of Switzerland, and is highly urbanized, with 26 academic medical centers. Therefore, most patients with HCC receive treatment in medical centers.

There were some limitations in this study. First, this is a retrospective study.

Second, the HCC registry data used in the study did not record extrahepatic comorbidities, and the severities of comorbidities are associated with survival and the treatment received. Also, we also did not review how many patients did not comply with the recommendations of the MDT. Moreover, the HCC registry data used only recorded the first-line therapy; therefore, we could not analyze the data regarding further treatments. In addition, sorafenib treatment has been reimbursed by the NHIA since August 2012 only for patients with PVTT (Vp3 or Vp4) [15] or EHS and Child-Pugh class A liver disease. Finally, we only enrolled patients who were diagnosed and managed at this hospital. Therefore, patients who were diagnosed at this hospital but received treatment at other hospital were excluded. That could explain the low percentage of patients who received BSC in our cohort.

The strengths of this study were as follows. First, we contacted every patient who was lost to follow-up by phone and checked the vital status of these patients by using the Cancers Screening and Tracing Information Integrated System for Taiwan (https://hosplab.hpa.gov.tw/CSTIIS/index.aspx). Therefore, we could make sure the vital status of every single patient enrolled in the present study. Second, this study investigated patients treated in a single liver transplant center. All of the patients received care from subspecialists and had their cases reviewed by an MDT. All of the treatment modalities for HCC were available to them and reimbursed by the NHIA. As such, variables that could modulate treatment decisions in clinical practice, including the availability of treatment procedures, expertise of the hospital, and financial constraints, could be excluded. Finally, since referral is not required in Taiwan, there was no referral bias in this study.

In conclusion, the differences in findings between the current study conducted in the East and the Italian study from the West were as follows. First, HBV is the leading etiology of chronic liver disease in Taiwan, whereas HCV is the leading etiology of chronic liver disease in Italy [9]. Second, EBRT and cytotoxic chemotherapy are among the treatment modalities used in patients with MVI and EHS in Taiwan [19], whereas these modalities are not recommended by the EASL guideline [17] and, therefore, were not used in the Italian study [9]. Third, referral is not required in Taiwan, whereas referral is required in most of the western countries.

## Supporting information

**S1 Data.**
(XLSX)

## Author Contributions

**Conceptualization:** Yi-Hao Yen, Tsung-Hui Hu.

**Data curation:** Yi-Hao Yen.

**Formal analysis:** Yi-Hao Yen.

**Funding acquisition:** Yi-Hao Yen.

**Investigation:** Yi-Hao Yen.

**Methodology:** Yi-Hao Yen.

**Supervision:** Yu-Fan Cheng, Jing-Houng Wang, Chih-Che Lin, Yen-Yang Chen, Chee-Chien Yong, Yueh-Wei Liu, Jen-Yu Cheng, Chien-Hung Chen, Tsung-Hui Hu.

**Validation:** Yi-Hao Yen.

**Writing – original draft:** Yi-Hao Yen.

**Writing – review & editing:** Yi-Hao Yen, Tsung-Hui Hu.

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
