## [Decision Letter · Decision Letter 0]

20 Jan 2020

PONE-D-19-35575

Real world clinical practice in treating advanced hepatocellular carcinoma: when East meets West

PLOS ONE

Dear Dr. Tsung-Hui Hu,

Thank you for submitting your manuscript to PLOS ONE. After careful consideration, we feel that it has merit but does not fully meet PLOS ONE’s publication criteria as it currently stands. Therefore, we invite you to submit a revised version of the manuscript that addresses the points raised during the review process.

We would appreciate receiving your revised manuscript within 60 days. To enhance the reproducibility of your results, we recommend that if applicable you deposit your laboratory protocols in protocols.io, where a protocol can be assigned its own identifier (DOI) such that it can be cited independently in the future. For instructions see: http://journals.plos.org/plosone/s/submission-guidelines#loc-laboratory-protocols

We look forward to receiving your revised manuscript.

Kind regards,

Gianfranco D. Alpini

Academic Editor

PLOS ONE

Journal Requirements:

2. Please modify the title to ensure that it meets PLOS’ guidelines (https://journals.plos.org/plosone/s/submission-guidelines#loc-title). In particular, the title should be "specific, descriptive, concise, and comprehensible to readers outside the field". In this case, the inclusion of the phrase "when East meets West" suggests that the study is comparing treatments between these two geographic areas. We would therefore recommend rephrasing the title as "Clinical treatments and outcomes of advanced hepatocellular carcinoma patients in an Asian medical center" (or similar).

Reviewers' comments:

Reviewer's Responses to Questions

**Comments to the Author**

1. Is the manuscript technically sound, and do the data support the conclusions?

Reviewer #1: Yes

Reviewer #2: Yes

2. Has the statistical analysis been performed appropriately and rigorously? 

Reviewer #1: Yes

Reviewer #2: Yes

3. Have the authors made all data underlying the findings in their manuscript fully available?

Reviewer #1: Yes

Reviewer #2: Yes

4. Is the manuscript presented in an intelligible fashion and written in standard English?

Reviewer #1: Yes

Reviewer #2: Yes

5. Review Comments to the Author

Reviewer #1: Yi-Hao Yen and coautours describe in this manuscript the outcome of a cohort of patients with HCC BCLC C in Taiwan. They described a good survival in stage C patients through the use of personalized management via MDT than that achieved by the one-size-fits-all approach used in clinical trials.

The topic is relevant, the study well conducted despite being a retrospective analysis, the analysis scientific sound.

I only have minor comments:

Among MVI patients, did author looked at whether OS was longer in those with peripheral than with central (portal trunk) MVI?

Methods: it should be reported the standard procedure of the MDT in their centre/s

Abstract: "We achieved better survival in BCLC C patients through the use of personalized management via a MDT approach." - please clarify with who, the comparison is performed

Discussion: 17.1% of the patients with MVI or EHS in this study underwent surgical resection - could you please report details on their outcomes

Discussion: "HBV is the leading etiology of chronic liver disease in Taiwan and the prevalence of non-cirrhotic HCC in Taiwan may be high [28], which may explain why there was a higher proportion of

patients with Child-Pugh class A liver functional reserve in our cohort" - how can you make this statement without reporting histological data?

Reviewer #2: In this paper authors aim to evaluate the real world clinical treatment and survival of BCLC C patients in an Asian cohort. 427 consecutive BCLC C patients diagnosed between were included. The median overall survival (OS) was 11.0 months in the whole cohort, which was longer than that reported in a previous phase III trial of sorafenib in advanced HCC patients conducted in Asia. The most frequent treatments were transcatheter arterial embolization (TAE) in the PS1 (45.8%) and PS2 patients (52.2%) and sorafenib (32.4%) in the macrovascular invasion (MVI) or extrahepatic spread (EHS). The independent prognostic factors were the PS, Child-Pugh class, MVI or EHS, alpha fetoprotein levels, and treatment type.

This paper is an interesting report of the real world multidisciplinary approach to therapy of advanced HCC in the east world, which, follow a recent well published paper from Italy with the same aim.

Major comments:

Please include limits of the paper. In particular authors should mention that this study cannot be compared with data of a registration clinical trial, like the one of sorafenib, as done. Moreover authors should highlight how the design of this paper is a limit per se.

Authors should details what is the novelty of the paper and differences with respect Ref. 6

Author should clearly explain the criteria for allocating people to the different therapeutic approaches, in particular when the decision differs with respect current international guidelines.

In the abstract, please specify that Sorafeninb is among the recommended therapy, and not the only one, since a new first line therapy has been approved.

6. PLOS authors have the option to publish the peer review history of their article (what does this mean?). If published, this will include your full peer review and any attached files.

Reviewer #1: Yes: marco carbone

Reviewer #2: No

---

## [Author Response · Author response to Decision Letter 0]

14 Feb 2020

Reviewer #1: Yi-Hao Yen and coauthors describe in this manuscript the outcome of a cohort of patients with HCC BCLC C in Taiwan. They described a good survival in stage C patients through the use of personalized management via MDT than that achieved by the one-size-fits-all approach used in clinical trials.

The topic is relevant, the study well conducted despite being a retrospective analysis, the analysis scientific sound.

I only have minor comments:

Among MVI patients, did author looked at whether OS was longer in those with peripheral than with central (portal trunk) MVI?

Response: Thank you so much for your comments. 

To assess the degree of MVI on the effect of OS, MVI patients were divided further according to the location/extent of vascular invasion. According to the Hong Kong Liver Cancer (HKLC) staging system [20], intrahepatic MVI was defined as tumor invasion of the intrahepatic branches of the portal vein or hepatic veins. Extrahepatic MVI was defined as tumor invasion of the main portal trunk or inferior vena cava. 

There were 321 patients with MVI or EHS in our study. Among these, 75 patients had MVI+EHS (Patients with both MVI and EHS), 51 had EHS alone (Patients with EHS and without MVI), and 195 had MVI alone (Patients with MVI and without EHS). For patients with MVI alone (n=195), according to the HKLC staging system [20], there were 73 with extrahepatic vascular invasion [i.e., central MVI (c-MVI)], 120 with intrahepatic vascular invasion [i.e., peripheral MVI (p-MVI)], and 2 with unclassified MVI (i.e., the location/extent of PVT was not mentioned in the image reports). The median OS was longer in patients with p-MVI (18.0 months; 95 % CI, 12.0-24.0) compared to those with c-MVI (6.0 months; 95 % CI, 3.0-7.0; p<0.001; Fig.3).

After patients with hepatic vein and/or inferior vena cava tumor invasion (n=48) were excluded, there are 58 patients with central (portal trunk) PVTT, 87 with peripheral PVTT, and 2 with unclassified PVTT. The median OS was longer in patients with p-MVI (14.1 months; 95 % CI, 9.0-22.0) compared to those with c-MVI (5.1 months; 95 % CI, 3.0-7.0; p=0.002; Fig.4).

Page 19-20. 

Methods: it should be reported the standard procedure of the MDT in their centre/s

Response: Thank you so much for your comments. 

The standard procedure of the MDT in our center

The MDT in our center includes hepatologists, radiologists, pathologists, transplant surgeons, surgical oncologists, radiation oncologists, medical oncologists, and nurses. The leader of our MDT is Yu-Fan Cheng. The MDT meetings are held every Tuesday, and all new HCC cases are discussed. Treatment experiences cases are discussed if the physician in charge requests. If MDT recommends BSC, the physician in charge will consult the palliative care professionals. If liver transplant is recommended, we refer these patients to liver transplant MDT for detailed discussion. 

Our MDT include two nurses who administer and coordinate the meetings. The general flow begins with a brief clinical presentation by the physician in charge, followed by images review, discussion, and decision. A standardized one-page form is used. This form records the relevant clinical and laboratory data that determine liver function, PS, the tumor stage and the consensus recommendations. The physician in charge informs patients of the MDT recommendations. The document of the MDT recommendations is kept in the patient’s medical chart. Page 10, last paragraph and page 11, first paragraph. 

Abstract: "We achieved better survival in BCLC C patients through the use of personalized management via a MDT approach." - please clarify with who, the comparison is performed

Response: Thank you so much for your comments. 

1. In our study, we followed the publication by Giannini et al. [1]. They reported that they achieved a median survival of 22.3 months in BCLC C patients using a patient-tailored management established by a MDT, which is remarkably longer than the one achieved with sorafenib in both randomized and post-marketing Western studies [2,3]. 

2. In our study, using patient-tailored management established by an MDT, we achieved a median survival of 11.0 months for the BCLC stage C patients in the real-world cohort, which is longer than the one achieved with sorafenib in a randomized controlled trial conducted in Asia [4].

3. However, I have deleted this paragraph because of another reviewer’s comments: In particular, the authors should mention that this study cannot be compared with data from a registration clinical trial, like the one of sorafenib, as done.

References: 

1. Giannini EG, Bucci L, Garuti F, Brunacci M, Lenzi B, Valente M, et al; Italian Liver Cancer (ITA.LI.CA) group. Patients with advanced hepatocellular carcinoma need a personalized management: A lesson from clinical practice. Hepatology. 2018;67:1784-1796

2. Llovet JM, Ricci S, Mazzaferro V, Hilgard P, Gane E, Blanc JF, et al. SHARP Investigators Study Group. Sorafenib in advanced hepatocellular carcinoma. N Engl J Med 2008;359:378-390.

3. Iavarone M, Cabibbo G, Piscaglia F, Zavaglia C, Grieco A, Villa E, et al. Field-practice study of sorafenib therapy for hepatocellular carcinoma: a prospective multicenter study in Italy. Hepatology. 2011;54:2055-2063

4. Cheng AL, Kang YK, Chen Z, Tsao CJ, Qin S, Kim JS, et al. Efficacy and safety of sorafenib in patients in the Asia-Pacific region with advanced hepatocellular carcinoma: A phase III randomised, double-blind, placebo-controlled trial. Lancet Oncol. 2009;10:25-34.

Discussion: 17.1% of the patients with MVI or EHS in this study underwent surgical resection - could you please report details on their outcomes

Response: Thank you so much for your comments. 

Fifty-five (17.1%) of the patients with MVI or EHS in this study underwent surgical resection. Among those, 2 had MVI and EHS, 5 had EHS alone, and 48 had MVI alone. The median overall survival was 67 months; 95% CI, 23-not available (not available indicates that the 95% CI survival has not yet been reached). Page 20, last paragraph. 

Discussion: "HBV is the leading etiology of chronic liver disease in Taiwan and the prevalence of non-cirrhotic HCC in Taiwan may be high [28], which may explain why there was a higher proportion of patients with Child-Pugh class A liver functional reserve in our cohort" - how can you make this statement without reporting histological data?

Response: Thank you so much for your comments. I have deleted this paragraph. 

Reviewer #2: In this paper authors aim to evaluate the real world clinical treatment and survival of BCLC C patients in an Asian cohort. 427 consecutive BCLC C patients diagnosed between were included. The median overall survival (OS) was 11.0 months in the whole cohort, which was longer than that reported in a previous phase III trial of sorafenib in advanced HCC patients conducted in Asia. The most frequent treatments were transcatheter arterial embolization (TAE) in the PS1 (45.8%) and PS2 patients (52.2%) and sorafenib (32.4%) in the macrovascular invasion (MVI) or extrahepatic spread (EHS). The independent prognostic factors were the PS, Child-Pugh class, MVI or EHS, alpha fetoprotein levels, and treatment type.

This paper is an interesting report of the real world multidisciplinary approach to therapy of advanced HCC in the east world, which, follow a recent well published paper from Italy with the same aim.

Major comments:

Please include limits of the paper. In particular authors should mention that this study cannot be compared with data of a registration clinical trial, like the one of sorafenib, as done. Moreover authors should highlight how the design of this paper is a limit per se.

Response: Thank you so much for your comments. 

1. I have deleted the paragraph regarding this study compared with data from a registration clinical trial, like the one of sorafenib. 

2. I have highlighted the way the design of this paper is a limitation per se. The abstract states that this is a retrospective cohort study that enrolled 427 consecutive BCLC stage C patients diagnosed between 2011 and 2017 by using the HCC registry data for our hospital. The patients and methods section states: This is a retrospective study using the last version of the Kaohsiung Chang Gung Memorial Hospital HCC registry data. The discussion states: There were some limitations in this study. First, this is a retrospective study.

Authors should details what is the novelty of the paper and differences with respect Ref. 6

Response: Thank you so much for your comments. 

1. A recent Italian study reported real world clinical practice results in treating naïve BCLC stage C HCC [9]. They found that the survival duration of BCLC stage C patients was improved through the use of personalized management via a multidisciplinary team (MDT) approach. However, the leading etiology of chronic liver disease in the European population is hepatitis C virus (HCV), and validation of the Italian study’s approach in other populations with different ethnic and clinical backgrounds, as well as in contexts with variations in the provision of systemic and local-regional therapies, is needed. We therefore investigated the real world clinical treatment and survival of BCLC stage C patients in an Asian medical center, most of whom were infected with hepatitis B virus (HBV). Page 5, last paragraph and page 6, first paragraph. 

2. Nearly 30% of the patients with MVI or EHS in our cohort underwent other treatments (i.e., systemic chemotherapy, HAIC, or EBRT). In contrast, none of the patients in the Italian study received these kinds of treatment [9]. In the west, although EBRT is being increasingly used in patients with MVI, it is not recommended by the EASL guideline [17]. In contrast, EBRT is recommended for patients with HCC and MVI in the Asia-Pacific and Taiwan guidelines [19, 26]. Meanwhile, HAIC is widely used for patients with HCC and PVTT in Japanese hospitals [27]. It is also one of the treatment modalities for patients with MVI recommended by the Taiwan guideline [19]. Page 29, last paragraph and page 30, first paragraph. 

3. Serper et al. reported that subspecialist care within 30 days of HCC diagnosis and review by an MDT were associated with reduced mortality compared with care provided by gastroenterologists in a community setting in a Veterans Administration cohort [28]. The current healthcare system in Taiwan, known as the National Health Insurance system, was instituted in 1995. The population coverage reached and has been maintained at or above 99% since 2004. Under the system, citizens are free to choose hospitals and physicians without referral. The NHIA offers reimbursement for all treatment modalities for HCC patients, as patients with cancers can apply a catastrophic illness card. As such, patients with HCC do not have to pay anything when they receive medical care related to HCC. Furthermore, the island of Taiwan has an area of 35,808 square kilometers, which is smaller than that of Switzerland, and is highly urbanized, with 26 academic medical centers. Therefore, most patients with HCC receive treatment in medical centers. Page 30, last paragraph and page 31, first paragraph. 

4. The strengths of this study were as follows. First, we contacted every patient who was lost to follow-up by phone and checked the vital status of these patients by using the Cancers Screening and Tracing Information Integrated System for Taiwan (https://hosplab.hpa.gov.tw/CSTIIS/index.aspx). Therefore, we could make sure the vital status of every single patient enrolled in the present study. Second, this study investigated patients treated in a single liver transplant center. All of the patients received care from subspecialists and had their cases reviewed by an MDT. All of the treatment modalities for HCC were available to them and reimbursed by the NHIA. As such, variables that could modulate treatment decisions in clinical practice, including the availability of treatment procedures, expertise of the hospital, and financial constraints, could be excluded. Finally, since referral is not required in Taiwan, there was no referral bias in this study. page 32, 2nd paragraph. 

5. In conclusion, the differences in findings between the current study conducted in the East and the Italian study from the West were as follows. First, HBV is the leading etiology of chronic liver disease in Taiwan, whereas HCV is the leading etiology of chronic liver disease in Italy [9]. Second, EBRT and cytotoxic chemotherapy are among the treatment modalities used in patients with MVI and EHS in Taiwan [19], whereas these modalities are not recommended by the EASL guideline [17] and, therefore, were not used in the Italian study [9]. Third, referral is not required in Taiwan, whereas referral is required in most of the western countries. Page 32, last paragraph and page 33, 1st paragraph. 

Author should clearly explain the criteria for allocating people to the different therapeutic approaches, in particular when the decision differs with respect current international guidelines.

Response: Thank you so much for your comments. 

Sorafenib treatment has been reimbursed by the National Health Insurance Administration (NHIA) in Taiwan since August 2012 for patients with portal vein tumor thrombus (PVTT) (Vp3 or Vp4) [15] or EHS and Child-Pugh class A liver disease. Therefore, BCLC C patients with PS1 alone, PS2 alone, Child-Pugh class B liver disease, peripheral PVTT (Vp1 or Vp2), hepatic veins tumor thrombus, inferior vena cava tumor thrombus and those diagnosed during 2011 to August 2012 were not reimbursed. 

For patients with PS1 alone or PS2 alone, the criteria for allocating patients to the different therapeutic approaches are as follows: 

1. Liver transplant is recommended as the first-line option for HCC within the University of California at San Francisco (UCSF) criteria but unsuitable for resection [16].

2. The general principle for surgical resection of HCC followed the recommendations of the European Association for the Study of the Liver (EASL) guidelines [17], which are based on a multi-parametric composite assessment of liver function, the extent of resection, the future liver remnant, PS, and the patient’s comorbidities. Resection is indicated in HCC patients with tumors located in one lobe of the liver [18]. 

3. RFA is indicated for patients with Child-Pugh class A or B liver disease and HCC within Milan criteria [18].

4. TACE or EBRT is indicated for patients with Child-Pugh class A or B liver disease and HCC within Milan criteria [10] in whom it is difficult to perform RFA because of medical comorbidities or tumor location [19].

5. TACE is indicated for patients with Child-Pugh class A or B liver disease and tumor burden at the BCLC stage B [19]. 

6. EBRT is indicated for patients with Child-Pugh class A or B liver disease and tumor burden at the BCLC stage B unsuitable to TACE [19]. 

Under most circumstances, patients with MVI or EHS were treated with sorafenib if they met the criteria of reimbursement. The criteria for allocating patients to the different therapeutic approaches are as follows:

1. RFA is indicated for patients with Child-Pugh class A or B liver disease and intrahepatic tumor burden within Milan criteria [10], with small metastatic lymph nodes or tiny lung nodules, misclassified as benign at the time of treatment. 

2. TAE/TACE is indicated for patients with Child-Pugh class A or B liver disease and intrahepatic tumor burden at the BCLC stage B, with small metastatic lymph nodes or tiny lung nodules, misclassified as benign at the time of treatment. 

3. Systemic chemotherapy or HAIC is indicated in patients with Child-Pugh class A liver disease and adequate hemogram data (white blood cell ≥ 4000/mL, platelet ≥ 100,000/mL) [22].

4. HAIC is indicated for patients with MVI [19]. 

5. Systemic chemotherapy is indicated for patients with EHS [19]. 

6. Resection is indicated in patients with HCC involving the ipsilateral portal vein. Resection is contraindicated in patients with HCC with contralateral portal vein or portal vein main trunk invasion [19].

7. Resection is indicated in patients with HCC involving ipsilateral hepatic vein. Resection is contraindicated in patients with HCC with right atrium or inferior vena cava involvement [19].

8. Resection is indicated in patients with HCC with extra-hepatic, single organ involvement. Resection is contraindicated in patients with HCCs with extra-hepatic, multiple organ involvement [19].

9. EBRT is indicated in patients with MVI and Child-Pugh class A or B liver disease [19].

Page 7, last paragraph to page 10, first paragraph. 

In the abstract, please specify that Sorafeninb is among the recommended therapy, and not the only one, since a new first line therapy has been approved.

Response: Thank you so much for your comments. I have corrected according to your comments: sorafeninb is one of the recommended therapies.

---

## [Decision Letter · Decision Letter 1]

20 Feb 2020

Real world clinical practice in treating advanced hepatocellular carcinoma: when East meets West

PONE-D-19-35575R1

Dear Dr. Tsung-Hui Hu,

We are pleased to inform you that your manuscript has been judged scientifically suitable for publication and will be formally accepted for publication once it complies with all outstanding technical requirements.

With kind regards,

Gianfranco D. Alpini

Academic Editor

PLOS ONE

Additional Editor Comments (optional):

Reviewers' comments:

Reviewer's Responses to Questions

**Comments to the Author**

1. If the authors have adequately addressed your comments raised in a previous round of review and you feel that this manuscript is now acceptable for publication, you may indicate that here to bypass the “Comments to the Author” section, enter your conflict of interest statement in the “Confidential to Editor” section, and submit your "Accept" recommendation.

Reviewer #2: All comments have been addressed

2. Is the manuscript technically sound, and do the data support the conclusions?

Reviewer #2: Yes

3. Has the statistical analysis been performed appropriately and rigorously? 

Reviewer #2: Yes

4. Have the authors made all data underlying the findings in their manuscript fully available?

Reviewer #2: Yes

5. Is the manuscript presented in an intelligible fashion and written in standard English?

Reviewer #2: Yes

6. Review Comments to the Author

Reviewer #2: All comments have been addressed by the authors. The manuscript has been significantly ameliorated after the revision.

7. PLOS authors have the option to publish the peer review history of their article (what does this mean?). If published, this will include your full peer review and any attached files.

Reviewer #2: No

---

## [Editor Report · Acceptance letter]

25 Feb 2020

PONE-D-19-35575R1 

Real world clinical practice in treating advanced hepatocellular carcinoma: when East meets West 

Dear Dr. Hu:

I am pleased to inform you that your manuscript has been deemed suitable for publication in PLOS ONE. Congratulations! Your manuscript is now with our production department. 

With kind regards,

on behalf of

Dr. Gianfranco D. Alpini 

Academic Editor

PLOS ONE